# Mistake Bounds for Binary Matrix Completion

**Mark Herbster**
University College London
Department of Computer Science
London WC1E 6BT, UK
m.herbster@cs.ucl.ac.uk

**Stephen Pasteris**
University College London
Department of Computer Science
London WC1E 6BT, UK
s.pasteris@cs.ucl.ac.uk

**Massimiliano Pontil**
Istituto Italiano di Tecnologia
16163 Genoa, Italy
and
University College London
Department of Computer Science
London WC1E 6BT, UK
m.pontil@cs.ucl.ac.uk

## Abstract

We study the problem of completing a binary matrix in an online learning setting. On each trial we predict a matrix entry and then receive the true entry. We propose a Matrix Exponentiated Gradient algorithm [1] to solve this problem. We provide a mistake bound for the algorithm, which scales with the *margin complexity* [2, 3] of the underlying matrix. The bound suggests an interpretation where each row of the matrix is a prediction task over a finite set of objects, the columns. Using this we show that the algorithm makes a number of mistakes which is comparable up to a logarithmic factor to the number of mistakes made by the Kernel Perceptron with an optimal kernel in hindsight. We discuss applications of the algorithm to predicting as well as the best biclustering and to the problem of predicting the labeling of a graph without knowing the graph in advance.

## 1 Introduction

We consider the problem of predicting *online* the entries in an $m \times n$ binary matrix $\boldsymbol{U}$. We formulate this as the following game: *nature* queries an entry $(i_1, j_1)$; the *learner* predicts $\hat{y}_1 \in \{-1, 1\}$ as the matrix entry; *nature* presents a label $y_1 = U_{i_1,j_1}$; *nature* queries the entry $(i_2, j_2)$; the *learner* predicts $\hat{y}_2$; and so forth. The learner's goal is to minimize the total number of mistakes $M = |\{t : \hat{y}_t \neq y_t\}|$. If nature is adversarial, the learner will always mispredict, but if nature is regular or simple, there is hope that a learner may make only a few mispredictions.

In our setting we are motivated by the following interpretation of matrix completion. Each of the $m$ rows represents a task (or binary classifier) and each of the $n$ columns is associated with an object (or input). A task is the problem of predicting the binary label of each of the objects. For a single task, if we were given a kernel matrix between the objects in advance we could then use the Kernel Perceptron algorithm to sequentially label the objects and this algorithm would incur $\mathcal{O}(1/\lambda^2)$ mistakes, where $\lambda$ is the margin of the best linear classifier in the inner product space induced by the kernel. Unfortunately, in our setup, we do not know a good kernel in advance. However, we will show that a remarkable property of our algorithm is that it enjoys, up to logarithmic factors, a mistake bound of $\mathcal{O}(1/\gamma^2)$ per task, where $\gamma$ is the largest possible margin (over the choice of the kernel) which is achieved on all tasks.

The problem of predicting online the labels of a finite set of objects under the assumption that the similarity between objects can be described by a graph was introduced in [4], building upon earlier work in the batch setting [5, 6]. In this and later research the common assumption is that two objects are similar if there is an edge in the graph connecting them and the aim is to predict well when there are few edges between objects with disagreeing labels. Lower bounds and an optimal algorithm (up to logarithmic factors) for this problem were given in [7, 8]. The problem of predicting well when the graph is unknown was previously addressed in [9, 10]. That research took the approach that when receiving a vertex to predict, edges local to that vertex were then revealed. In this paper we take a different approach - the graph structure is never revealed to the learner. Instead, we have a number of tasks over the same unknown graph, and the hope is to perform comparably to the case in which the graph in known in advance.

The general problem of matrix completion has been studied extensively in the batch statistical i.i.d. setting, see for example [11, 12, 13] and references therein. These studies are concerned either with Rademacher bounds or statistical oracle inequalities, both of which are substantially different from the focus of the present paper. In the online mistake-bound setting a special form of matrix completion was previously considered as the problem of learning a binary relation [14, 15] (see Section 5). In a more general online setting, with minimal assumptions on the loss function [16, 17] bounded the regret of the learner in terms of the *trace-norm* of the underlying matrix. Instead our bounds are with respect to the *margin complexity* of the matrix. As a result, although our bounds have a more restricted applicability they have the advantage that they become non-trivial after only $\tilde{\Theta}(n)$ matrix entries[1] are observed as opposed to the required $\tilde{\Theta}(n^{3/2})$ in [16] and $\tilde{\Theta}(n^{7/4})$ in [17]. The notion of margin complexity in machine learning was introduced in [2] where it was used to the study the learnability of concept classes via linear embeddings and further studied in [3], where it was linked to the $\gamma_2$ norm. Here we adopt the terminology in [11] and refer to the $\gamma_2$ norm as the max-norm. The margin complexity seems to be a more natural parameter as opposed to the trace-norm for the 0-1 loss as it only depends on the signs of the underlying comparator matrix. To the best of our knowledge the bounds contained herein are the first online matrix completion bounds in terms of the margin complexity.

To obtain our results, we use an online matrix multiplicative weights algorithm, e.g., see [1, 18, 17, 19] and references therein. These kinds of algorithms have been applied in a number of learning scenarios, including online PCA [20], online variance minimization [21], solving SDPs [18], and online prediction with switching sequences [22]. These algorithms update a new hypothesis matrix on each trial by trading off fidelity to the previous hypothesis and the incorporation of the new label information. The tradeoff is computed as an approximate spectral regularization via the quantum relative entropy (see [1, Section 3.1]). The particular matrix multiplicative weights algorithm we apply is Matrix Winnow [19]; we adapt this algorithm and its mistake bound analysis for our purposes via selection of comparator, threshold, and appropriate "progress inequalities."

The paper is organized as follows. In Section 2 we introduce basic notions used in the paper. In Section 3 we present our algorithm and derive a mistake bound, also comparing it to related bounds in the literature. In Section 4 we observe that our algorithm is able to exploit matrix structure to perform comparably to the Kernel Perceptron with the best kernel known in advance. Finally, in Section 5 we discuss the example of biclustered matrices, and argue that our bound is optimal up to a polylogarithmic factor. The appendix contains proofs of the results only stated in the main body of the paper, and other auxiliary results.

## 2 Preliminaries

We denote the set of the first $m$ positive integers as $\mathbb{N}_m = \{1, \ldots, m\}$. We denote the inner product of vectors $\boldsymbol{x}, \boldsymbol{w} \in \mathbb{R}^n$ as $\langle \boldsymbol{x}, \boldsymbol{w} \rangle = \sum_{i=1}^{n} x_i w_i$ and the norm as $|\boldsymbol{w}| = \sqrt{\langle \boldsymbol{w}, \boldsymbol{w} \rangle}$. We let $\mathbb{R}^{m \times n}$ be the set of all $m \times n$ real-valued matrices. If $\boldsymbol{X} \in \mathbb{R}^{m \times n}$ then $\boldsymbol{X}_i$ denotes the $i$-th $n$-dimensional row vector and the $(i, j)$ entry in $\boldsymbol{X}$ is $X_{ij}$. The trace of a square matrix $\boldsymbol{X} \in \mathbb{R}^{n \times n}$ is $\text{Tr}(\boldsymbol{X}) = \sum_{i=1}^{n} X_{ii}$. The trace norm of a matrix $\boldsymbol{X} \in \mathbb{R}^{m \times n}$ is $\|\boldsymbol{X}\|_1 = \text{Tr}(\sqrt{\boldsymbol{X}^\top \boldsymbol{X}})$, where $\sqrt{\cdot}$ indicates the unique positive square root of a positive semi-definite matrix. For every matrix $\boldsymbol{U} \in \{-1, 1\}^{m \times n}$, we define $\text{SP}(\boldsymbol{U}) = \{\boldsymbol{V} \in \mathbb{R}^{m \times n} : \forall_{ij} V_{ij} U_{ij} > 0\}$, the set of matrices which

are sign consistent with $\boldsymbol{U}$. We also define $\mathrm{SP}^1(\boldsymbol{U}) = \{\boldsymbol{V} \in \mathbb{R}^{m \times n} : \forall_{ij} V_{ij} U_{ij} \geq 1\}$, that is the set of matrices which are sign consistent to $\boldsymbol{U}$ with a margin of at least one.

The max-norm (or $\gamma_2$ norm [3]) of a matrix $\boldsymbol{U} \in \mathbb{R}^{m \times n}$ is defined by the formula

$$\|\boldsymbol{U}\|_{\max} := \inf_{\boldsymbol{PQ}^\top = \boldsymbol{U}} \left\{ \max_{1 \leq i \leq m} |\boldsymbol{P}_i| \max_{1 \leq j \leq n} |\boldsymbol{Q}_j| \right\}, \tag{1}$$

where the infimum is over all matrices $\boldsymbol{P} \in \mathbb{R}^{m \times k}$ and $\boldsymbol{Q} \in \mathbb{R}^{n \times k}$ and every integer $k$. The *margin complexity* of a matrix $\boldsymbol{U} \in \mathbb{R}^{m \times n}$ is

$$\mathrm{mc}(\boldsymbol{U}) := \inf_{\boldsymbol{PQ}^\top \in \mathrm{SP}(\boldsymbol{U})} \max_{ij} \frac{|\boldsymbol{P}_i||\boldsymbol{Q}_j|}{|\langle \boldsymbol{P}_i, \boldsymbol{Q}_j \rangle|}.$$

This quantity plays a central role in the analysis of our algorithm. If we interpret the rows of $\boldsymbol{U}$ as $m$ different binary classification tasks, and the columns as a finite set of objects which we wish to label, the "min-max" margin with respect to an embedding is smallest of the $m$ maximal margins over the tasks. The quantity $1/\mathrm{mc}(\boldsymbol{U})$ is then the maximum "min-max" margin with respect to all possible embeddings. Specifically, the rows of matrix $\boldsymbol{P}$ represent the "weights" of the binary classifiers and the rows of matrix $\boldsymbol{Q}$ the "input vectors" associated with the objects. The quantity $\frac{|\langle \boldsymbol{P}_i, \boldsymbol{Q}_j \rangle|}{|\boldsymbol{P}_i||\boldsymbol{Q}_j|}$ is the margin of the $i$-th classifier on the $j$-th input. Observe that margin complexity depends only on the sign pattern of the matrix and not the magnitudes. The margin complexity is equivalently $\mathrm{mc}(\boldsymbol{U}) = \min_{\boldsymbol{V} \in \mathrm{SP}^1(\boldsymbol{U})} \|\boldsymbol{V}\|_{\max}$, see e.g., [3, Lemma 3.1].

In our online setting we are concerned with predicting an *(example) sequence* $((i_1, j_1), y_1), \dots, ((i_T, j_T), y_T) \in (\mathbb{N}_m \times \mathbb{N}_n) \times \{-1, 1\}$. A sequence must be *consistent*, that is, given examples $((i, j), y)$ and $((i', j'), y')$ if $(i, j) = (i', j')$ then $y = y'$. We define the set of sign-consistent matrices with a sequence $\mathcal{S}$ as $\mathrm{cons}(\mathcal{S}) := \{\boldsymbol{M} \in \mathbb{R}^{m \times n} : 0 < y M_{ij}, ((i, j), y) \in \mathcal{S}\}$. We extend the notion of margin complexity to sequences via $\mathrm{mc}(\mathcal{S}) := \inf_{\boldsymbol{U} \in \mathrm{cons}(\mathcal{S})} \mathrm{mc}(\boldsymbol{U})$.

The number of margin violations in a sequence $\mathcal{S}$ at complexity $\gamma$ is defined to be,

$$\mathrm{merr}(\mathcal{S}, \gamma) := \inf_{\boldsymbol{PQ}^\top \in \mathrm{cons}(\mathcal{S})} \left| \left\{ ((i, j), y) \in \mathcal{S} : \frac{|\boldsymbol{P}_i||\boldsymbol{Q}_j|}{|\langle \boldsymbol{P}_i, \boldsymbol{Q}_j \rangle|} > \frac{1}{\gamma} \right\} \right|. \tag{2}$$

In particular, note that $\mathrm{merr}(\mathcal{S}, \gamma) = 0$ if $\gamma \leq \frac{1}{\mathrm{mc}(\mathcal{S})}$.

Finally, we introduce the following quantity, which plays a central role in the amortized analysis of our algorithm.

**Definition 2.1.** *The quantum relative entropy of symmetric positive semidefinite square matrices $\boldsymbol{A}$ and $\boldsymbol{B}$ is*

$$\Delta(\boldsymbol{A}, \boldsymbol{B}) := \mathrm{Tr}(\boldsymbol{A} \log(\boldsymbol{A}) - \boldsymbol{A} \log(\boldsymbol{B}) + \boldsymbol{B} - \boldsymbol{A}).$$

## 3 Algorithm and Analysis

Algorithm 1 presents an adaptation of the Matrix Exponentiated Gradient algorithm [1, 17, 18, 19] to our setting. This algorithm is a matrix analog of the Winnow algorithm [19]; we refer to the above papers for more insights into this family of algorithms.

The following theorem provides a mistake bound for the algorithm.

**Theorem 3.1.** *The number of mistakes, $M$, on sequence $\mathcal{S}$ made by the Algorithm 1 with parameter $0 < \gamma \leq 1$ is upper bounded by*

$$M \leq c \left[ (m + n) \log(m + n) \frac{1}{\gamma^2} + \mathrm{merr}(\mathcal{S}, \gamma) \right], \tag{3}$$

*where $c = 1/(3 - e) \leq 3.55$ and the quantity $\mathrm{merr}(\mathcal{S}, \gamma)$ is given in equation* (2).

*Proof.* Given $\boldsymbol{U} \in \mathbb{R}^{m \times n}$, let $\boldsymbol{P} \in \mathbb{R}^{m \times k}$ and $\boldsymbol{Q} \in \mathbb{R}^{n \times k}$ be such that $\boldsymbol{PQ}^\top = \boldsymbol{U}$. For every $i \in \mathbb{N}_m$, we denote by $\boldsymbol{P}_i$ the $i$-th row vector of $\boldsymbol{P}$ and for every $j \in \mathbb{N}_n$, we denote by $\boldsymbol{Q}_j$ the $j$-th row vector of $\boldsymbol{Q}$. We construct the $(m + n) \times k$ matrix

$$\boldsymbol{R} := \mathrm{diag}\left( \frac{1}{|\boldsymbol{P}_1|}, \dots, \frac{1}{|\boldsymbol{P}_m|}, \frac{1}{|\boldsymbol{Q}_1|}, \dots, \frac{1}{|\boldsymbol{Q}_n|} \right) \begin{bmatrix} \boldsymbol{P} \\ \boldsymbol{Q} \end{bmatrix}$$

**Algorithm 1** Predicting a binary matrix.

**Parameters:** Learning rate $0 < \gamma \le 1$.

**Initialization:** $\boldsymbol{W}^{(0)} \leftarrow \frac{\boldsymbol{I}}{(m+n)}$, where $\boldsymbol{I}$ is the $(m+n) \times (m+n)$ identity matrix.

**For** $t = 1, \ldots, T$

- Get pair $(i_t, j_t) \in \mathbb{N}_m \times \mathbb{N}_n$.
- Define $\boldsymbol{X}^{(t)} := \frac{1}{2}(\boldsymbol{e}_{i_t} + \boldsymbol{e}_{m+j_t})(\boldsymbol{e}_{i_t} + \boldsymbol{e}_{m+j_t})^\top$, where $\boldsymbol{e}_k$ is the $k$-th basis vector of $\mathbb{R}^{m+n}$.
- Predict

$$\hat{y}_t = \begin{cases} 1 & \text{if } \mathrm{Tr}(\boldsymbol{W}^{(t-1)}\boldsymbol{X}^{(t)}) \ge \frac{1}{m+n}, \\ -1 & \text{otherwise.} \end{cases}$$

- Receive label $y_t \in \{-1, 1\}$ and if $\hat{y}_t \ne y_t$ update

$$\boldsymbol{W}^{(t)} \leftarrow \exp\left(\log\left(\boldsymbol{W}^{(t-1)}\right) + \frac{\gamma}{2}(y_t - \hat{y}_t)\boldsymbol{X}^{(t)}\right).$$

and construct $\tilde{\boldsymbol{U}} := (\frac{1}{m+n})\boldsymbol{R}\boldsymbol{R}^\top$. Define matrix $\boldsymbol{X}^{(t)} := \frac{1}{2}(\boldsymbol{e}_{i_t} + \boldsymbol{e}_{m+j_t})(\boldsymbol{e}_{i_t} + \boldsymbol{e}_{m+j_t})^\top$, where $\boldsymbol{e}_k$ is the $k$-th basis vector of $\mathbb{R}^{m+n}$.

Note that $\mathrm{Tr}(\boldsymbol{X}^{(t)}) = 1$, $\mathrm{Tr}(\tilde{\boldsymbol{U}}) = 1$ (since every row of $\boldsymbol{R}$ is normalized) and

$$\begin{aligned}
\mathrm{Tr}(\tilde{\boldsymbol{U}}\boldsymbol{X}^{(t)}) &= \frac{1}{n+m} \mathrm{Tr}((\boldsymbol{R}\boldsymbol{R}^\top)\frac{1}{2}(\boldsymbol{e}_{i_t} + \boldsymbol{e}_{m+j_t})(\boldsymbol{e}_{i_t} + \boldsymbol{e}_{m+j_t})^\top) \\
&= \frac{1}{2(n+m)}(\boldsymbol{e}_{i_t} + \boldsymbol{e}_{m+j_t})^\top \boldsymbol{R}\boldsymbol{R}^\top (\boldsymbol{e}_{i_t} + \boldsymbol{e}_{m+j_t}) \\
&= \frac{1}{2(n+m)}(\boldsymbol{R}^\top(\boldsymbol{e}_{i_t} + \boldsymbol{e}_{m+j_t}))^\top(\boldsymbol{R}^\top(\boldsymbol{e}_{i_t} + \boldsymbol{e}_{m+j_t})) \\
&= \frac{1}{2(n+m)}\left(\frac{\boldsymbol{P}_{i_t}}{|\boldsymbol{P}_{i_t}|} + \frac{\boldsymbol{Q}_{j_t}}{|\boldsymbol{Q}_{j_t}|}\right)\left(\frac{\boldsymbol{P}_{i_t}}{|\boldsymbol{P}_{i_t}|} + \frac{\boldsymbol{Q}_{j_t}}{|\boldsymbol{Q}_{j_t}|}\right)^\top \\
&= \frac{1}{(n+m)}\left(1 + \frac{\langle \boldsymbol{P}_{i_t}, \boldsymbol{Q}_{j_t}\rangle}{|\boldsymbol{P}_{i_t}||\boldsymbol{Q}_{j_t}|}\right).
\end{aligned}$$

For a trial $t$ we say there is a *margin violation* if $\frac{|\boldsymbol{P}_{i_t}||\boldsymbol{Q}_{j_t}|}{|\langle\boldsymbol{P}_{i_t},\boldsymbol{Q}_{j_t}\rangle|} > \frac{1}{\gamma}$. Let $M^-$ denote the number of mistakes made in trials with margin violations and let $M^+$ denote the number of mistakes made in trials without margin violations.

From Lemma A.3 in the appendix we have

$$\Delta(\tilde{\boldsymbol{U}}, \boldsymbol{W}^{(t-1)}) - \Delta(\tilde{\boldsymbol{U}}, \boldsymbol{W}^{(t)}) \ge \frac{\gamma}{2}(y_t - \hat{y}_t)\mathrm{Tr}(\tilde{\boldsymbol{U}}\boldsymbol{X}^{(t)}) + \left(1 - e^{\frac{\gamma}{2}(y_t - \hat{y}_t)}\right)\mathrm{Tr}(\boldsymbol{W}^{(t-1)}\boldsymbol{X}^{(t)}),$$

then substituting in the above we have that

$$\begin{aligned}
\Delta(\tilde{\boldsymbol{U}}, \boldsymbol{W}^{(t-1)}) - \Delta(\tilde{\boldsymbol{U}}, \boldsymbol{W}^{(t)}) &\ge \frac{\gamma}{2}(y_t - \hat{y}_t)\frac{1}{n+m}\left(1 + \frac{\langle\boldsymbol{P}_{i_t},\boldsymbol{Q}_{j_t}\rangle}{|\boldsymbol{P}_{i_t}||\boldsymbol{Q}_{j_t}|}\right) \\
&\quad + \left(1 - e^{\frac{\gamma}{2}(y_t - \hat{y}_t)}\right)\mathrm{Tr}(\boldsymbol{W}^{(t-1)}\boldsymbol{X}^{(t)}).
\end{aligned}$$

To further simplify the above we use Lemma A.4 presented in the appendix, which gives

$$\Delta(\tilde{\boldsymbol{U}}, \boldsymbol{W}^{(t-1)}) - \Delta(\tilde{\boldsymbol{U}}, \boldsymbol{W}^{(t)}) \ge \begin{cases} (c'-1)\frac{1}{n+m}\gamma^2, & \text{if there is a margin violation}, \\ c'\frac{1}{n+m}\gamma^2, & \text{otherwise.} \end{cases}$$

where $c' = 3 - e$.

Using a telescoping sum, this gives

$$
\begin{aligned}
\Delta(\tilde{U}, W^{(0)}) &\geq \Delta(\tilde{U}, W^{(0)}) - \Delta(\tilde{U}, W^{(T)}) \geq M^+ c' \frac{1}{n+m}\gamma^2 + M^-(c'-1)\frac{1}{n+m}\gamma^2 \\
&= (c'M^+ - (1-c')M^-)\frac{1}{n+m}\gamma^2
\end{aligned}
$$

and hence

$$
M^+ \leq \frac{1}{c' \frac{1}{n+m}\gamma^2}\Delta(\tilde{U}, W^{(0)}) + \frac{1-c'}{c'}M^- \ .
$$

We conclude that

$$
M = M^+ + M^- \leq \frac{1}{c' \frac{1}{n+m}\gamma^2}\Delta(\tilde{U}, W^{(0)}) + \frac{1}{c'}M^- \ .
$$

We also have that

$$
\begin{aligned}
\Delta(\tilde{U}, W^{(0)}) &= \mathrm{Tr}(\tilde{U}\log(\tilde{U})) - \mathrm{Tr}(\tilde{U}\log(W^{(0)})) + \mathrm{Tr}(W^{(0)}) - \mathrm{Tr}(\tilde{U}) \\
&= \mathrm{Tr}(\tilde{U}\log(\tilde{U})) - \mathrm{Tr}(\tilde{U}\log(W^{(0)})) + 1 - 1 \\
&= \mathrm{Tr}(\tilde{U}\log(\tilde{U})) - \mathrm{Tr}(\tilde{U}\log(W^{(0)})) \ .
\end{aligned}
$$

Write the eigen-decomposition of $\tilde{U}$ as $\sum_{i=1}^{m+n} \lambda_i \boldsymbol{\alpha}_i \boldsymbol{\alpha}_i^T$. Now we have $\sum_{i=1}^{m+n} \lambda_i = \mathrm{Tr}(\tilde{U}) = 1$ so all eigenvalues $\lambda_i$ are in the range $[0, 1]$ meaning $\log(\lambda_i) \leq 0$ so $\lambda_i \log(\lambda_i) < 0$ which are the eigenvalues of $\tilde{U}\log(\tilde{U})$ meaning that $\mathrm{Tr}(\tilde{U}\log(\tilde{U})) \leq 0$. Also, $\log(W^{(0)}) = \log(\frac{1}{n+m})I$ so $\tilde{U}\log(W^{(0)}) = \log(\frac{1}{n+m})\tilde{U}$ and hence $-\mathrm{Tr}(\tilde{U}\log(W^{(0)})) = -\log(\frac{1}{n+m})\mathrm{Tr}(\tilde{U}) = \log(m+n)$. So by the above we have

$$
\Delta(\tilde{U}, W^{(0)}) \leq \log(m+n)
$$

and hence putting together we get

$$
M \leq \frac{m+n}{c'\gamma^2}\log(m+n) + \frac{1}{c'}M^- \ .
$$

$\square$

Observe that in the simplifying case when we have no margin errors ($\mathrm{merr}(\mathcal{S}, \gamma) = 0$) and the learning rate is $\gamma := \frac{1}{\mathrm{mc}(\mathcal{S})}$ we have that the number of mistakes of Algorithm 1 is bounded by $\tilde{\mathcal{O}}((n+m)\,\mathrm{mc}^2(\mathcal{S}))$. More generally although the learning rate is fixed in advance, we may use a "doubling trick" to avoid the need to tune the $\gamma$.

**Corollary 3.2.** *For any value of $\gamma^*$ the number of mistakes $M$ made by the following algorithm:*

DOUBLING ALGORITHM:

*Set $\kappa \leftarrow \sqrt{2}$ and loop over*

> *1. Run Algorithm 1 with $\gamma = \frac{1}{\kappa}$ until it has made $\lceil 2c(m+n)\log(m+n)\kappa^2\rceil$ mistakes*
> *2. Set $\kappa \leftarrow \kappa\sqrt{2}$*

*is upper bounded by*

$$
M \leq 12c\left[(m+n)\log(m+n)\frac{1}{(\gamma^*)^2} + \mathrm{merr}(\mathcal{S}, \gamma^*)\right],
$$

*with $c = 1/(3 - e) \approx 3.55$.*

See the appendix for a proof. We now compare our bound to other online learning algorithms for matrix completion. The algorithms of [16, 17] address matrix completion in a significantly more general setting. Both algorithms operate with weak assumptions on the loss function, while our algorithm is restricted to the 0–1 loss (mistake counting). Those papers present regret bounds, whereas we apply the stronger assumption that there exists a consistent predictor. As a regret bound is not possible for a deterministic predictor with the 0–1 loss, we compare Theorem 3.1 to their

bound when their algorithm is allowed to predict $\hat{y} \in [-1, 1]$ and uses absolute loss. For clarity in our discussion we will assume that $m \in \Theta(n)$.

Under the above assumptions, the regret bound in [17, Corollary 7] becomes $2\sqrt{\|\boldsymbol{U}\|_1 (m+n)^{1/2} \log(m+n)T}$. For simplicity we consider the simplified setting in which each entry is predicted, that is $T = mn$; then absorbing polylogarithmic factors, their bound is $\tilde{\mathcal{O}}(n^{5/4}\|\boldsymbol{U}\|_1^{\frac{1}{2}})$. From Theorem 3.1 we have a bound of $\tilde{\mathcal{O}}(n \operatorname{mc}^2(\boldsymbol{U}))$. Using [11, Theorem 10], we may upper bound the margin complexity in terms of the trace norm,

$$\operatorname{mc}(\boldsymbol{U}) \leq 3 \min_{\boldsymbol{V} \in \operatorname{SP}^1(\boldsymbol{U})} \|\boldsymbol{V}\|_1^{\frac{1}{3}} \leq 3\|\boldsymbol{U}\|_1^{\frac{1}{3}} \,. \tag{4}$$

Substituting this into Theorem 3.1 our bound is $\tilde{\mathcal{O}}(n\|\boldsymbol{U}\|_1^{\frac{2}{3}})$. Since the trace norm may be bounded as $n \leq \|\boldsymbol{U}\|_1 \leq n^{3/2}$, both bounds become vacuous when $\|\boldsymbol{U}\|_1 = n^{3/2}$, however if the trace norm is bounded away from $n^{3/2}$, the bound of Theorem 3.1 is smaller by a polynomial factor. An aspect of the bounds which this comparison fails to capture is the fact that since [17, Corollary 7] is a regret bound it will degrade more smoothly under adversarial noise than Theorem 3.1.

The algorithm in [16] is probabilistic and the regret bound is of $\tilde{\mathcal{O}}(\|\boldsymbol{U}\|_1\sqrt{n})$. Unlike [17], the setting of [16] is transductive, that is each matrix entry is seen only once, and thus less general. If we use the upper bound from [11, Theorem 10] as in the discussion of [17] then [16] improves uniformly on our bound and the bound in [17]. However, using this upper bound oversimplifies the comparison as $1 \leq \operatorname{mc}^2(\boldsymbol{U}) \leq n$ while $n \leq \|\boldsymbol{U}\|_1 \leq n^{3/2}$ for $\boldsymbol{U} \in \{-1, 1\}^{m \times n}$. In other words we have been very conservative in our comparison; the bound (4) may be loose and our algorithm may often have a much smaller bound. A specific example is provided by the class of $(k, \ell)$-biclustered matrices (see also the discussion in Section 5 below) where $\operatorname{mc}^2(\boldsymbol{U}) \leq \min(k, \ell)$, in which case bound becomes nontrivial after $\tilde{\Theta}(\min(k, \ell) n)$ examples while the bounds in [16] and [17] become nontrivial after at least $\tilde{\Theta}(n^{3/2})$ and $\tilde{\Theta}(n^{7/4})$ examples, respectively.

With respect to computation our algorithm on each trial requires a single eigenvalue decomposition of a PSD matrix, whereas the algorithm of [17] requires multiple eigenvalue decompositions per trial. Although [16] does not discuss the complexity of their algorithm beyond the fact that it is polynomial, in [17] it is conjectured that it requires at a minimum $\Theta(n^4)$ time per trial.

## 4 Comparison to the Best Kernel Perceptron

In this section, we observe that Algorithm 1 has a mistake bound that is comparable to Novikoff's bound [23] for the Kernel Perceptron with an optimal kernel in hindsight. To explain our observation, we interpret the rows of matrix $\boldsymbol{U}$ as $m$ different binary classification tasks, and the columns as a finite set of objects which we wish to label; think for example of users/movies matrix in recommendation systems. If we solve the tasks independently using a Kernel Perceptron algorithm, we will make $O(1/\gamma^2)$ mistakes per task, where $\gamma$ is the largest margin of a consistent hypothesis. If every task has a margin larger than $\gamma$ we will make $O(m/\gamma^2)$ mistakes in total. This algorithm and the parameter $\gamma$ crucially depend on the kernel used: if there exists a kernel which makes $\gamma$ large for all (or most of) the tasks, then the Kernel Perceptron will incur a small number of mistakes on all (or most of) the tasks. We now argue that our bound mimics this "oracle", without knowing in advance the kernel. Without loss of generality, we assume $m \geq n$ (otherwise apply the same reasoning below to matrix $\boldsymbol{U}^\top$). In this scenario, Theorem 3.1 upper bounds the number of mistakes as

$$O\left(\frac{m \log m}{\gamma^2}\right)$$

where $\gamma$ is chosen so that $\operatorname{merr}(\mathcal{S}, \gamma) = 0$. To further illustrate our idea, we define the *task complexity* of a matrix $\boldsymbol{U} \in \mathbb{R}^{m \times n}$ as

$$\tau(\boldsymbol{U}) = \min\left\{h(\boldsymbol{V}) : \boldsymbol{V} \in \operatorname{SP}^1(\boldsymbol{U})\right\}$$

where

$$h(\boldsymbol{V}) = \inf_{\boldsymbol{K} \succ 0} \max_{1 \leq i \leq m} \boldsymbol{V}_i \boldsymbol{K}^{-1} \boldsymbol{V}_i^\top \max_{1 \leq j \leq n} K_{jj} \,. \tag{5}$$

Note that the quantity $\boldsymbol{V}_i \boldsymbol{K}^{-1} \boldsymbol{V}_i^\top \max_{1 \leq j \leq n} K_{jj}$ is exactly the bound in Novikoff's Theorem on the number of mistakes of the Kernel Perceptron on the $i$-th task with kernel $\boldsymbol{K}$. Hence the quantity

$h(\boldsymbol{V})$ represents the best upper bound on the number of mistakes made by a Kernel Perceptron on the worst (since we take the maximum over $i$) task.

**Proposition 4.1.** *For every $\boldsymbol{U} \in \mathbb{R}^{m \times n}$, it holds that $\mathrm{mc}^2(\boldsymbol{U}) = \tau(\boldsymbol{U})$.*

*Proof.* The result follows by Lemma A.6 presented in the appendix and by the formula $\mathrm{mc}(\boldsymbol{U}) = \min_{\boldsymbol{V} \in \mathrm{SP}^1(\boldsymbol{U})} \|\boldsymbol{V}\|_{\max}$, see, e.g., [3, Lemma 3.1]. $\qquad\square$

Returning to the interpretation of the bound in Theorem 3.1, we observe that if no more than $r$ out of the $m$ tasks have margin smaller than a threshold $\lambda$ then in Algorithm 1 setting parameter $\gamma = \lambda$, Theorem 3.1 gives a bound of

$$O\left(\frac{(m-r)\log m}{\lambda^2} + rn\right).$$

Thus we essentially "pay" linearly for every object in a difficult task. Since we assume $n \leq m$, provided $r$ is small the bound is "robust" to the presence of bad tasks.

We specialize the above discussion to the case that each of the $m$ tasks is a binary labeling of an unknown underlying connected graph $\mathcal{G} := (\mathcal{V}, \mathcal{E})$ with $n$ vertices and assume that $m \geq n$. We let $\boldsymbol{U} \in \{-1, 1\}^{m \times n}$ be the matrix, the rows of which are different binary labelings of the graph. For every $i \in \mathbb{N}_m$, we interpret $\boldsymbol{U}_i$, the $i$-th row of matrix $\boldsymbol{U}$, as the $i$-th labeling of the graph and let $\Phi_i$ be the corresponding cutsize, namely, $\Phi_i := |\{(j, j') \in \mathcal{E} : U_{ij} \neq U_{ij'}\}|$ and define $\Phi_{\max} := \max_{1 \leq i \leq m} \Phi_i$. In order to apply Theorem 3.1, we need to bound the margin complexity of $\boldsymbol{U}$. Using the above analysis (Proposition 4.1), this quantity is upper bounded by

$$\mathrm{mc}^2(\boldsymbol{U}) \leq \max_{1 \leq i \leq m} \boldsymbol{U}_i \boldsymbol{K}^{-1} \boldsymbol{U}_i^\top \max_{1 \leq j \leq n} K_{jj}. \tag{6}$$

We choose the kernel $\boldsymbol{K} := \boldsymbol{L}^+ + (R\,\mathbf{1}\mathbf{1}^T)$, where $\boldsymbol{L}$ is the graph Laplacian of $\mathcal{G}$, the vector $\mathbf{1}$ has all components equal to one, and $R = \max_j L_{jj}^+$. Since the graph is connected then $\mathbf{1}$ is the only eigenvector of $\boldsymbol{L}$ with zero eigenvalue. Hence $\boldsymbol{K}$ is invertible and $\boldsymbol{K}^{-1} = \boldsymbol{L} + (R\,\mathbf{1}\mathbf{1}^T)^+ = \boldsymbol{L} + (R\,n\,\frac{1}{\sqrt{n}}\mathbf{1}\mathbf{1}^T\frac{1}{\sqrt{n}})^+ = \boldsymbol{L} + \frac{1}{Rn^2}\mathbf{1}\mathbf{1}^T$. Then using the formula $\Phi_i = \frac{1}{4}\boldsymbol{U}_i \boldsymbol{L}\boldsymbol{U}_i^\top$ we obtain from (6) that

$$\mathrm{mc}^2(\boldsymbol{U}) \leq \max_{1 \leq i \leq m} \left(4\Phi_i + \frac{1}{R}\right) R.$$

Theorem 3.1 then gives a bound of $M \leq O\left((1 + \Phi_{\max}R)\,m \log m\right)$. The quantity $R$ may be further upper bounded by the graph resistance diameter, see for example [24].

## 5   Biclustering and Near Optimality

The problem of learning a $(k, \ell)$-binary-biclustered matrix, corresponds to the assumption that the row indices and column indices represent $k$ and $\ell$ distinct object types and that there exists a binary relation on these objects which determines the matrix entry. Formally we have the following

**Definition 5.1.** *The class of $(k, \ell)$-binary-biclustered matrices is defined as*

$$\mathbb{B}_{k,\ell}^{m,n} = \{\boldsymbol{U} \in \mathbb{R}^{m \times n} : \boldsymbol{r} \in \mathbb{N}_k^m, \boldsymbol{c} \in \mathbb{N}_\ell^n, \boldsymbol{F} \in \{-1, 1\}^{k \times \ell},\ U_{ij} = F_{r_i c_j},\ i \in \mathbb{N}_m, j \in \mathbb{N}_n\}.$$

The intuition is that a matrix is $(k, \ell)$-biclustered if after a permutation of the rows and columns the resulting matrix is a $k \times \ell$ grid of rectangles and all entries in a given rectangle are either $1$ or $-1$. The problem of determining a $(k, \ell)$-biclustered matrix with a minimum number of "violated" entries given a subset of entries was shown to be NP-hard in [25]. Thus although we do not give an algorithm that provides a biclustering, we provide a bound in terms of the best consistent biclustering.

**Lemma 5.2.** *If $\boldsymbol{U} \in \mathbb{B}_{k,\ell}^{m,n}$ then $\mathrm{mc}^2(\boldsymbol{U}) \leq \min(k, \ell)$.*

*Proof.* We use Proposition 4.1 to upper bound $\mathrm{mc}^2(\boldsymbol{U})$ by $h(\boldsymbol{U})$, where the function $h$ is given in equation (5). We further upper bound $h(\boldsymbol{U})$ by choosing a kernel matrix in the underlying optimization problem. By Definition 5.1, there exists $\boldsymbol{r} \in \mathbb{N}_k^m, \boldsymbol{c} \in \mathbb{N}_\ell^n$ and $\boldsymbol{F} \in \{-1, 1\}^{k \times \ell}$

such that $U_{ij} = F_{r_i c_j}$, for every $i \in \mathbb{N}_m$ and every $j \in \mathbb{N}_n$. Then we choose the kernel matrix $\boldsymbol{K} = (K_{jj'})_{1 \leq j,j' \leq n}$ such that

$$K_{jj'} := \delta_{c_j c'_j} + \epsilon \delta_{jj'}$$

One verifies that $\boldsymbol{U}_i \boldsymbol{K}^{-1} \boldsymbol{U}_i^\top \leq \ell$ for every $i \in \{1, \dots, m\}$, hence by taking the limit for $\epsilon \to 0$ Proposition 4.1 gives that $\mathrm{mc}^2(\boldsymbol{U}) \leq \ell$. By the symmetry of our construction we can swap $\ell$ with $k$, giving the bound. $\qquad\square$

Using this lemma with Theorem 3.1 gives us the following upper bound on the number of mistakes.

**Corollary 5.3.** *The number of mistakes of Algorithm 1 applied to sequences generated by a $(k, \ell)$-binary-biclustered matrix is upper bounded by $\mathcal{O}(\min(k, \ell)(m+n)\log(m+n))$.*

A special case of the setting in this corollary was first studied in the mistake bound setting in [14]. In [15] the bound was improved and generalized to include robustness to noise (for simplicity we do not compare in the noisy setting). In both papers the underlying assumption is that there are $k$ distinct row types and no restrictions on the number of columns thus $\ell = n$. In this case they obtained an upper bound of $kn + \min(\frac{m^2}{2e}\log_2 e, m\sqrt{3n \log_2 k})$. Comparing the two bounds we can see that when $k < n^{\frac{1}{2}-\epsilon}$ the bound in Corollary 5.3 improves over [15, Corollary 1] by a polynomial factor and on other hand when $k \geq n^{\frac{1}{2}}$ we are no worse than a polylogarithmic factor.

We now establish that the mistake bound (3) is tight up to a poly-logarithmic factor.

**Theorem 5.4.** *Given an online algorithm $\mathcal{A}$ that predicts the entries of a matrix $\boldsymbol{U} \in \{-1, 1\}^{m \times n}$ and given an $\ell \in \mathbb{N}_n$ there exists a sequence $\mathcal{S}$ constructed by an adversary with margin complexity $\mathrm{mc}(\mathcal{S}) \leq \sqrt{\ell}$. On this sequence the algorithm $\mathcal{A}$ will make at least $\ell \times m$ mistakes.*

See the appendix for a proof.

# 6 Conclusion

In this paper, we presented a Matrix Exponentiated Gradient algorithm for completing the entries of a binary matrix in an online learning setting. We established a mistake bound for this algorithm, which is controlled by the margin complexity of the underlying binary matrix. We discussed improvements of the bound over related bounds for matrix completion. Specifically, we noted that our bound requires fewer examples before it becomes non-trivial, as compared to the bounds in [16, 17]. Here we require only $\tilde{\Theta}(m+n)$ examples as opposed to the required $\tilde{\Theta}((m+n)^{3/2})$ in [16] and $\tilde{\Theta}((m+n)^{7/4})$, respectively. Thus although our bound is more sensitive to noise, it captures structure more quickly in the underlying matrix. When interpreting the rows of the matrix as binary tasks, we argued that our algorithm performs comparably (up to logarithmic factors) to the Kernel Perceptron with the optimal kernel in retrospect. Finally, we highlighted the example of completing a biclustered matrix and noted that this is instrumental in showing the optimality of the algorithm in Theorem 5.4.

We observed that Algorithm 1 has a per trial computational cost which is smaller than currently available algorithms for matrix completion with online guarantees. In the future it would be valuable to study if improvements in this computation are possible by exploiting the special structure in our algorithm. Furthermore, it would be very interesting to study a modification of our analysis to the case in which the tasks (rows of matrix $\boldsymbol{U}$) grow over time, a setting which resembles the lifelong learning frameworks in [26, 27].

**Acknowledgements.** We wish to thank the anonymous reviewers for their useful comments. This work was supported in part by EPSRC Grants EP/P009069/1, EP/M006093/1, and by the U.S. Army Research Laboratory and the U.K. Defence Science and Technology Laboratory and was accomplished under Agreement Number W911NF-16-3-0001. The views and conclusions contained in this document are those of the authors and should not be interpreted as representing the official policies, ether expressed or implied, of the U.S. Army Research Laboratory, the U.S. Government, the U.K. Defence Science and Technology Laboratory or the U.K. Government. The U.S. and U.K. Governments are authorized to reproduce and distribute reprints for Government purposes notwithstanding any copyright notation herein.

## Footnotes

[1] For simplicity we assume $m \in \Theta(n)$.

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
