[Supplementary Material]

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

}^{(0)}) &= \operatorname{Tr}(\tilde{\boldsymbol{U}} \log(\tilde{\boldsymbol{U}})) - \operatorname{Tr}(\tilde{\boldsymbol{U}} \log(\boldsymbol{W}^{(0)})) + \operatorname{Tr}(\boldsymbol{W}^{(0)}) - \operatorname{Tr}(\tilde{\boldsymbol{U}}) \\
&= \operatorname{Tr}(\tilde{\boldsymbol{U}} \log(\tilde{\boldsymbol{U}})) - \operatorname{Tr}(\tilde{\boldsymbol{U}} \log(\boldsymbol{W}^{(0)})) + 1 - 1 \\
&= \operatorname{Tr}(\tilde{\boldsymbol{U}} \log(\tilde{\boldsymbol{U}})) - \operatorname{Tr}(\tilde{\boldsymbol{U}} \log(\boldsymbol{W}^{(0)})) .
\end{aligned}
$$

Write the eigen-decomposition of $\tilde{\boldsymbol{U}}$ as $\sum_{i=1}^{m+n} \lambda_i \boldsymbol{\alpha}_i \boldsymbol{\alpha}_i^T$. Now we have $\sum_{i=1}^{m+n} \lambda_i = \operatorname{Tr}(\tilde{\boldsymbol{U}}) = 1$ so all eigenvalues $\lambda_i$ are in the range $[0, 1]$ meaning $\log(\lambda_i) \leq 0$ so $\lambda_i \log(\lambda_i) < 0$ which are the eigenvalues of $\tilde{\boldsymbol{U}} \log(\tilde{\boldsymbol{U}})$ meaning that $\operatorname{Tr}(\tilde{\boldsymbol{U}} \log(\tilde{\boldsymbol{U}})) \leq 0$. Also, $\log(\boldsymbol{W}^{(0)}) = \log(\frac{1}{n+m})\boldsymbol{I}$ so $\tilde{\boldsymbol{U}} \log(\boldsymbol{W}^{(0)}) = \log(\frac{1}{n+m})\tilde{\boldsymbol{U}}$ and hence $-\operatorname{Tr}(\tilde{\boldsymbol{U}} \log(\boldsymbol{W}^{(0)})) = -\log(\frac{1}{n+m}) \operatorname{Tr}(\tilde{\boldsymbol{

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

 collect some auxiliary results. The first result is the well known Golden-Thompson Inequality, whose proof can be found, for example, in [28].

**Lemma A.1.** *For any symmetric matrices $\boldsymbol{A}$ and $\boldsymbol{B}$ we have that*

$$\text{Tr}(\exp(\boldsymbol{A}+\boldsymbol{B})) \leq \text{Tr}(\exp(\boldsymbol{A})\exp(\boldsymbol{B})). \tag{7}$$

The next result is taken from [1].

**Lemma A.2.** *If $\boldsymbol{A} \succ 0$ with eigenvalues in $[0,1]$ and $a \in \mathbb{R}$, then*

$$(1 - e^a)\boldsymbol{A} \preceq \boldsymbol{I} - \exp(a\boldsymbol{A}).$$

The next lemma is useful for the analysis of Matrix Winnow, see [1, 19, 22].

**Lemma A.3.**

$$\Delta(\tilde{\boldsymbol{U}}, \boldsymbol{W}^{(t-1)}) - \Delta(\tilde{\boldsymbol{U}}, \boldsymbol{W}^{(t)}) \geq \frac{\gamma}{2}(y_t - \hat{y}_t)\,\text{Tr}(\tilde{\boldsymbol{U}}\boldsymbol{X}^{(t)}) + \left(1 - e^{\frac{\gamma}{2}(y_t - \hat{y}_t)}\right)\text{Tr}(\boldsymbol{W}^{(t-1)}\boldsymbol{X}^{(t)})$$

*Proof.* We observe that

$$
\begin{aligned}
\Delta(\tilde{\boldsymbol{U}}, \boldsymbol{W}^{(t-1)}) - \Delta(\tilde{\boldsymbol{U}}, \boldsymbol{W}^{(t)}) &= \text{Tr}(\tilde{\boldsymbol{U}}\log\boldsymbol{W}^{(t)} - \tilde{\boldsymbol{U}}\log\boldsymbol{W}^{(t-1)}) + \text{tr}\,\boldsymbol{W}^{(t-1)} - \text{tr}\,\boldsymbol{W}^{(t)} \\[2mm]
&= \frac{\gamma}{2}(y_t - \hat{y}_t)\,\text{Tr}(\tilde{\boldsymbol{U}}\boldsymbol{X}^{(t)}) + \text{tr}\,\boldsymbol{W}^{(t-1)} - \text{tr}\left(e^{\log\boldsymbol{W}^{(t-1)} + \frac{\gamma}{2}(y_t - \hat{y}_t)\boldsymbol{X}^{(t)}}\right) \\[2mm]
&\geq \frac{\gamma}{2}(y_t - \hat{y}_t)\,\text{Tr}(\tilde{\boldsymbol{U}}\boldsymbol{X}^{(t)}) + \text{tr}\,\boldsymbol{W}^{(t-1)} - \text{tr}\left(e^{\log\boldsymbol{W}^{(t-1)}}e^{\frac{\gamma}{2}(y_t - \hat{y}_t)\boldsymbol{X}^{(t)}}\right) \\[2mm]
&= \frac{\gamma}{2}(y_t - \hat{y}_t)\,\text{Tr}(\tilde{\boldsymbol{U}}\boldsymbol{X}^{(t)}) + \text{tr}\left(\boldsymbol{W}^{(t-1)}\left(\boldsymbol{I} - e^{\frac{\gamma}{2}(y_t - \hat{y}_t)\boldsymbol{X}^{(t)}}\right)\right) \\[2mm]
&\geq \frac{\gamma}{2}(y_t - \hat{y}_t)\,\text{Tr}(\tilde{\boldsymbol{U}}\boldsymbol{X}^{(t)}) + (1 - e^{\frac{\gamma}{2}(y_t - \hat{y}_t)})\,\text{Tr}(\boldsymbol{W}^{(t-1)}\boldsymbol{X}^{(t)})
\end{aligned}
$$

where the second equality follows by the update formula in Algorithm 1, the first inequality follows by Golden-Thompson Inequality (Lemma A.1), and the last inequality follows by Lemma A.2. □

The next lemma further analyzes the quantity $\Delta(\tilde{\boldsymbol{U}}, \boldsymbol{W}^{(t-1)}) - \Delta(\tilde{\boldsymbol{U}}, \boldsymbol{W}^{(t)})$ distinguishing between the case in which there is a margin violation or not.

**Lemma A.4.** *Let $c' = 3 - e$. If $y_t \neq \hat{y}_t$, it holds that*

$$\Delta(\tilde{\boldsymbol{U}}, \boldsymbol{W}^{(t-1)}) - \Delta(\tilde{\boldsymbol{U}}, \boldsymbol{W}^{(t)}) \geq \begin{cases} (c' - 1)\frac{1}{n+m}\gamma^2, & \text{if there is a margin violation}, \\[2mm] c'\frac{1}{n+m}\gamma^2, & \text{otherwise}. \end{cases}$$

*Proof.* If $y_t = 1, \hat{y}_t = -1$ and there is not a margin violation then

$$
\begin{aligned}
\Delta(\tilde{\boldsymbol{U}}, \boldsymbol{W}^{(t-1)}) - \Delta(\tilde{\boldsymbol{U}}, \boldsymbol{W}^{(t)}) &\geq \gamma\frac{1}{n+m}\left(1 + \frac{\langle\boldsymbol{P}_{i_t}, \boldsymbol{Q}_{j_t}\rangle}{|\boldsymbol{P}_{i_t}||\boldsymbol{Q}_{j_t}|}\right) + \frac{1}{n+m}(1 - e^{\gamma}) \\[2mm]
&\geq \frac{1}{n+m}(\gamma + \gamma^2 + 1 - e^{\gamma}) \\[2mm]
&\geq c'\frac{1}{n+m}\gamma^2.
\end{aligned}
$$

And if $y_t = -1$, $\hat{y}_t = 1$ and there is not a margin violation then

$$\Delta(\tilde{U}, W^{(t-1)}) - \Delta(\tilde{U}, W^{(t)}) \geq -\gamma \frac{1}{n+m} \left(1 + \frac{\langle P_{i_t}, Q_{j_t} \rangle}{|P_{i_t}||Q_{j_t}|}\right) + \frac{1}{n+m}(1 - e^{-\gamma})$$

$$\geq \frac{1}{n+m}(-\gamma + \gamma^2 + 1 - e^{-\gamma})$$

$$\geq c' \frac{1}{n+m}\gamma^2.$$

If $y_t = 1$, $\hat{y}_t = -1$ and there is a margin violation then

$$\Delta(\tilde{U}, W^{(t-1)}) - \Delta(\tilde{U}, W^{(t)}) \geq \gamma \frac{1}{n+m} \left(1 + \frac{\langle P_{i_t}, Q_{j_t} \rangle}{|P_{i_t}||Q_{j_t}|}\right) + \frac{1}{n+m}(1 - e^{\gamma})$$

$$\geq \frac{1}{n+m}(\gamma + \gamma 0 + 1 - e^{\gamma})$$

$$= \frac{1}{n+m}(\gamma + 1 - e^{\gamma})$$

$$\geq (c' - 1)\frac{1}{n+m}\gamma^2.$$

And if $y_t = -1$, $\hat{y}_t = 1$ and there is a margin violation then

$$\Delta(\tilde{U}, W^{(t-1)}) - \Delta(\tilde{U}, W^{(t)}) \geq -\gamma \frac{1}{n+m} \left(1 + \frac{\langle P_{i_t}, Q_{j_t} \rangle}{|P_{i_t}||Q_{j_t}|}\right) + \frac{1}{n+m}(1 - e^{-\gamma})$$

$$\geq -\frac{1}{n+m}(\gamma - \gamma 0 - 1 + e^{-\gamma})$$

$$= \frac{1}{n+m}(-\gamma + 1 - e^{-\gamma})$$

$$\geq (c' - 1)\frac{1}{n+m}\gamma^2.$$

So on a mistaken trial $t$ without a margin violation we have

$$\Delta(\tilde{U}, W^{(t-1)}) - \Delta(\tilde{U}, W^{(t)}) \geq c' \frac{1}{n+m}\gamma^2$$

and on a mistaken trial $t$ with a margin violation we have

$$\Delta(\tilde{U}, W^{(t-1)}) - \Delta(\tilde{U}, W^{(t)}) \geq (c' - 1)\frac{1}{n+m}\gamma^2.$$

The constant $c'$ is chosen via Lemma A.5 below. $\qquad\square$

**Lemma A.5.** *For every $x \in [0, 1]$ it holds that*

$$\min(x^2 - x + 1 - e^{-x}, x^2 + x + 1 - e^x) \geq (3 - e)x^2.$$

*Proof.* Let $f(x) := x^2 - x + 1 - e^{-x} - (3 - e)x^2$ and $g(x) := x^2 + x + 1 - e^x - (3 - e)x^2$.

We first show that $f(x) \geq 0$. Taking a derivative we have that $f'(x) := e^{-x} + (2e - 4)x$. Since $f'(0) = 0$ and $f'(x)$ is strictly increasing for $x \geq 0$ this implies that $f(x) \geq 0$ for $x \geq 0$.

Second, we show that $g(x) \geq 0$. Taking derivatives we have that $g'(x) = 1 + (2e - 4)x - e^x$ and $g''(x) = (2e - 4) - e^x$. Since $g''$ is strictly decreasing, $g'$ is concave and has at most two zero-crossings. Therefore $g$ has at most two critical points. One critical point is at 0 and the other is at an $x^* \in [0.6, 0.8]$. Since $g''(0)$ is positive, 0 is a minimum of $g$ and since $g''(x) < 0$ for $0.6 < x$, $x^*$ is a maximum of $g$. Thus $g$ is unimodal in $[0, 1]$ with minimum values at $g(0) = g(1) = 0$. $\quad\square$

The following provides an alternative formulation of the max-norm.

**Lemma A.6.** *For every $V \in \mathbb{R}^{m \times n}$, we have that $\|V\|_{\max}^2 = h(V)$.*

*Proof.* Let $f(\boldsymbol{P}, \boldsymbol{Q}) = \max\limits_{1 \le i \le m} |\boldsymbol{P}_i|^2 \max\limits_{1 \le j \le n} |\boldsymbol{Q}_j|^2$. Then we have that

$$\|\boldsymbol{V}\|_{\max}^2 = \inf_{\boldsymbol{P}\boldsymbol{Q}^\top = \boldsymbol{V}} f(\boldsymbol{P}, \boldsymbol{Q}) \tag{8}$$

where the infimum is over all real matrices $\boldsymbol{P} \in \mathbb{R}^{m \times k}$, $\boldsymbol{Q} \in \mathbb{R}^{n \times k}$ and every integer $k$. For any $\boldsymbol{K} \succ 0$, set $\boldsymbol{P} = \boldsymbol{V}\sqrt{\boldsymbol{K}^{-1}}$ and $\boldsymbol{Q} = \sqrt{\boldsymbol{K}}$. Note that $\boldsymbol{P}\boldsymbol{Q}^\top = \boldsymbol{V}$ and, for every $i \in \mathbb{N}_m$, $|\boldsymbol{P}_i|^2 = \boldsymbol{V}_i \boldsymbol{K}^{-1} \boldsymbol{V}_i^\top$. Moreover, for any $j \in \mathbb{N}_n$, it holds $|\boldsymbol{Q}_j|^2 = \boldsymbol{K}_{jj}$. This shows that every feasible point in the optimization problem (5) maps to a feasible point for the optimization problem (8) and that the two objective functions take the same value at these points. Hence, we have shown, for every $\boldsymbol{V} \in \mathbb{R}^{m \times n}$, that

$$\|\boldsymbol{V}\|_{\max}^2 \le h(\boldsymbol{V}). \tag{9}$$

To show the reverse inequality, let $k \in \mathbb{N}$, let $\boldsymbol{P} \in \mathbb{R}^{m \times k}$ and $\boldsymbol{Q} \in \mathbb{R}^{n \times k}$ be a solution of the optimization problem (8). Set $\boldsymbol{K} = \boldsymbol{Q}\boldsymbol{Q}^\top + \epsilon\boldsymbol{I}$ for $\epsilon > 0$. Then for every $j \in \mathbb{N}_n$, $\boldsymbol{K}_{jj} = |\boldsymbol{Q}_j|^2 + \epsilon$ and, for every $i \in \mathbb{N}_m$, we have that

$$\boldsymbol{V}_i \boldsymbol{K}^{-1} \boldsymbol{V}_i^\top = \boldsymbol{e}_i^\top \boldsymbol{V} \boldsymbol{K}^{-1} \boldsymbol{V}^\top \boldsymbol{e}_i = \boldsymbol{e}_i^\top \boldsymbol{P} \boldsymbol{Q}^\top \boldsymbol{K}^{-1} \boldsymbol{Q} \boldsymbol{P}^\top \boldsymbol{e}_i \le \boldsymbol{P}_i \boldsymbol{P}_i^\top = |\boldsymbol{P}_i|^2$$

where we used the fact that $\boldsymbol{Q}^\top \left(\boldsymbol{Q}\boldsymbol{Q}^\top + \epsilon\boldsymbol{I}\right)^{-1} \boldsymbol{Q} \prec \boldsymbol{I}$. Thus

$$\max_{1 \le i \le m} \boldsymbol{V}_i \boldsymbol{K}^{-1} \boldsymbol{V}_i^\top \max_{1 \le j \le n} \boldsymbol{K}_{jj} < \max_{1 \le i \le m} |\boldsymbol{P}_i|^2 \max_{1 \le j \le n} \left(|\boldsymbol{Q}_j|^2 + \epsilon\right)$$

and taking the limit of $\epsilon \to 0$, we conclude that $h(\boldsymbol{V}) \le \|\boldsymbol{V}\|_{\max}^2$. This inequality, combined with inequality (9) proves the result. $\qquad\square$

*Proof of Corollary 3.2.* For any $\gamma > 0$ define $\mathcal{A}(\gamma) := c(m+n)\log(m+n)\frac{1}{\gamma^2}$, $\mathcal{B}(\gamma) := c\,\mathrm{merr}(\mathcal{S}, \gamma)$, and $\mathcal{M}(\gamma) := \mathcal{A}(\gamma) + \mathcal{B}(\gamma)$ which is equal to the mistake bound for Algorithm 1 with learning rate $\gamma$. Let $\eta$ be such that for all $\gamma \le \eta$ we have $\mathcal{A}(\gamma) \ge \mathcal{B}(\gamma)$ and for all $\gamma > \eta$ we have $\mathcal{A}(\gamma) \le \mathcal{B}(\gamma)$ which is defined and positive since $\mathcal{A}(\cdot)$ is continuous and monotonic decreasing (and limiting to 0) and $\mathcal{B}(\cdot)$ is monotonic non-decreasing (with $\mathcal{B}(0) = 0$). Since $\mathcal{A}(\cdot)$ is continuous let $\epsilon$ be such that $\epsilon > 0$ and $\mathcal{A}(\eta + \epsilon) \ge \mathcal{A}(\eta) - \frac{1}{2}$. For $\gamma \le \eta + \epsilon$ we have $\mathcal{M}(\gamma) \ge \mathcal{A}(\gamma) \ge \mathcal{A}(\eta + \epsilon)$ and for $\gamma \ge \eta + \epsilon$ we have $\mathcal{M}(\gamma) \ge \mathcal{B}(\gamma) \ge \mathcal{B}(\eta + \epsilon) \ge \mathcal{A}(\eta + \epsilon)$ so in either case we have $\mathcal{M}(\gamma) \ge \mathcal{A}(\eta + \epsilon) \ge \mathcal{A}(\eta) - \frac{1}{2}$.

Let $z$ be the minimum integer power of $\sqrt{2}$ that is greater than or equal to $1/\eta$. i.e. $z := \min\{\sqrt{2}^i : i \in \mathbb{N}, \sqrt{2}^i \ge 1/\eta\}$. Since $1/z \le \eta$ we have $\mathcal{B}(1/z) \le \mathcal{B}(\eta) \le \mathcal{A}(\eta) \le \mathcal{A}(1/z)$ so $\mathcal{M}(1/z) = \mathcal{A}(1/z) + \mathcal{B}(1/z) \le 2\mathcal{A}(1/z) = 2c(m+n)\log(m+n)z^2$ and hence the doubling algorithm above terminates at some $\kappa \le z$. The total number of mistakes, $M$, made by doubling algorithm above is then bounded above by

$$(2\mathcal{A}(2^{-1/2}) + 1) + (2\mathcal{A}(2^{-2/2}) + 1) + (2\mathcal{A}(2^{-3/2}) + 1) + \cdots + (2\mathcal{A}(1/z) + 1)$$
$$\le 3\mathcal{A}(2^{-1/2}) + 3\mathcal{A}(2^{-2/2}) + \cdots + 3\mathcal{A}(1/z)$$
$$\le 3c(m+n)\log(m+n)(2 + 2^2 + \cdots z^2)$$
$$\le 6c(m+n)\log(m+n)z^2 - 6$$
$$= 6\mathcal{A}(1/z) - 6.$$

We also have $z \le \frac{\sqrt{2}}{\eta}$ so $\mathcal{A}(1/z) \le 2\mathcal{A}(\eta)$ and putting together we get $M \le 12\mathcal{A}(\eta) - 6$ so since, by above, $\mathcal{A}(\eta) \le \mathcal{M}(\gamma) + \frac{1}{2}$ for all $\gamma > 0$ we're done. $\qquad\square$

*Proof of Theorem 5.4.* Observe that for the set of $(k, \ell)$-biclustered $m \times n$ matrices we have that,

$$k \times \ell \le \text{VC-dimension}(\mathbb{B}_{k,\ell}^{m,n})$$

as each of $k \times \ell$ "tiles" may be labeled independently of the others. Thus there exists an adversary that may force $m \times \ell$ mistakes via a constructed sequence $\mathcal{S}$ that is consistent with some $(m, \ell)$-biclustered matrix $\boldsymbol{U}$. By Lemma 5.2 we have that $\mathrm{mc}^2(\boldsymbol{U}) \le \min(m, \ell)$ and hence $\mathrm{mc}(\mathcal{S}) \le \sqrt{\ell}$. $\qquad\square$