[Reviews · NeurIPS 2016]

Reviewer 1

Summary

In this paper, the authors propose a Matrix Exponentiated Gradient algorithm to solve the problem of completing a binary matrix in an online learning setting. They provide a mistake bound for the algorithm, which scales with the margin complexity of the underlying matrix and is comparable up to a logarithmic factor to the number of mistakes made by the Kernel Perceptron with an optimal kernel in hindsight. The paper also discusses applications of the algorithm to predicting as well as the best biclustering and to the problem of predicting the labeling of a graph without knowing the graph in advance.

Qualitative Assessment

The work is well motivated. The author proposes a new algorithm, and gives an upper bound utilizing margin complexity. Compared with other algorithms, it shows that the upper bound of the proposed algorithm has certain advantages in some cases. The paper discusses the example of biclustered matrices. I think that the paper is relatively complete in the structure. My main problem with the paper relates to the analysis of the regret bound, the author claims that they have a bound of O ̃(n 〖mc〗^2 (U)) from Theorem 3.1, but there is lack of more details. I hope the author can explain how to obtain it concisely. Minor issues: The manuscript needs to be cleaned up, in particular the writing. Line 155 I do not know how to get the bound (2A(2^(-1/2) )+1)+(2A(2^(-2/2) )+1)+(2A(2^(-3/2) )+1)+⋯+(2A(1/ζ)+1)? Explain it more detail! Line 228 I do not clear how to get K^(-1)? Explain it more detail! Besides, what is the meaning about “mc” in Line 168? I hope the authors can make a detailed explanation of the above issues and revise these errors.

Confidence in this Review

2-Confident (read it all; understood it all reasonably well)


Reviewer 2

Summary

The manuscript considers the problem of predicting binary matrix entries in the online setting. The authors provide a mistake bound for the familiar matrix exponentiated gradient algorithm, where the bound scales with a notion of margin complexity of the ground-truth matrix. The novelty of the paper lies in the analysis; both the problem setting and the algorithm itself have been previously studied. The worst-case bounds offer some improvements over what can be deduced from existing results in some settings, but it is not clear how it plays out in the average case (or if this makes a difference in practice).

Qualitative Assessment

The problem of completing a binary matrix in the online learning setting has been studied before with fairly strong recovery guarantees in some settings (tight up to logarithmic factors, given lower bounds) [17]. The present paper proposes to use the familiar matrix exponentiated gradient algorithm for online binary matrix prediction, and provides a general mistake bound that scales with the margin complexity of the ground-truth matrix. The proposed algorithm is efficient and can be implemented to scale, compared to some existing solutions. The paper is well-written and analysis is presented clearly. However, the proposed algorithm and the bound is restricted to 0-1 loss. Furthermore, it assumes that there exists a consistent predictor (that makes no mistakes), which limits the applicability of the proposed framework; typically in online learning settings, one provides a regret bound. It is nice to see that the general mistake bound gives a reasonably tight bound for the problem of (k,l)-biclustering. On the other hand, the comparison to independent kernel perceptron (in the sense of best algorithm in hindsight) is not that interesting. In particular, the resulting bound scales with the number of tasks m. The bounds for the margin complexity in many places are pessimistic, which makes the bounds look uninteresting in some cases. The only exception being the (k,l)-biclustering problem where the bounds are reasonably tight. In contrast, approaches such as in [17], like in the case of standard (offline) matrix completion, the bounds scale with the complexity of some class of matrices (such as low-trace norm matrices). It would have been good to see some experimental simulations in order to give some insight into what happens in practice, but that is sorely missing from the paper. As a minor point, it would be good to highlight (or re-organize) the lower bound part presented in the end of Section 5 . As it stands, it is likely to be missed by readers. (I've read the author response).

Confidence in this Review

2-Confident (read it all; understood it all reasonably well)


Reviewer 3

Summary

The paper concerns completing binary matrix in the online manner: in each trial, the learner predicts the label in {-1,1}, the nature reveals the outcome and the learner suffers 0/1 loss. The interpretation is that each row is a task, and each column is an object, and the complexity of the problem is measures in terms of the margin complexity of the matrix. It is assumed that the problem is consistent in the sense that the entry seen multiple times alway receives the same label. The proposed algorithm to solve this problem is a version of the Matrix Exponentiated Gradient algorithm: a mistake bound in terms of margin complexity is provided, and it is shown that the algorithm makes a number of mistakes which is comparable up to a logarithmic factor to the number of mistakes made by the Kernel Perceptron with an optimal kernel in hindsight. Some applications are discussed: predicting as well as the best biclustering, and predicting the labeling of a graph without knowing the graph in advance.

Qualitative Assessment

The paper is clearly written and easy to follow. The technical quality is sounds and the proofs seem correct (except one issue in the proof of Lemma A.4, mentioned below). I am not entirely sure, whether there are any underlying assumptions on the data apart from the fact that the labels related to a given matrix entry are consistent (i.e., the same)? The authors claim that their setting is less general than in the previous papers on this topic, as they assume that there exists an underlying consistent predictor. Given the labels are consistent for each entry, is it sufficient to say that a consistent predictor exists? If so, I do not find this setting to be very restrictive, as in the real-life applications of matrix completion we often observe a given entry only once. I would be glad to get some feedback from the authors on this question. While the basic mistake bound for matrix exponentiated gradient algorithm makes use of the analysis the previous work on MEG, the embedding of the comparators (sign-consistent matrices) to the space of positive matrices, and the outcomes to the space of symmetric matrices looks very interesting and original. I also liked the interpretation of the comparator class in terms of linear predictors with any weights over any features for each object, which permits competing with the Kernel Perceptron with an optimal kernel in hindsight. One drawback is computational complexity of the algorithm, which can hinder the use of the algorithm in practical applications. I think, however, that the computational complexity could be somewhat improved by making use of the fact that the observed outcome matrices have rank one. Then, the eigenvalue decomposition of the exponent in the definition of W^(t) (Algorithm 1) can be maintained and updated using a faster algorithm for rank-one update of the eigenvalues and eigenvectors. I think the paper brings significant contribution to matrix completion and online learning, and thus I am in favor of accepting it for publication. Minor remarks: - Lemma A.4: How is Tr(W^(t-1) X^(t)) bounded in the lemma when \hat{y} = -1? It seems to lower-bounded by 1/(n+m), while according to the Algorithm it is below 1/(n+m) when \hat{y} = -1. - I think that part of the proof of Theorem 3.1, lines 129-135, is a quite standard bound on quantum relative entropy for matrices with unit trace (see, e.g., [1]), so it is probably not necessary to present it in full detail. - l. 88-89: "the quantity 1/ mc(U) represents the margin of the worst linear classifier in the best embedded space" -- why is it the worst linear classifier -- from the definition, it looks as if the inverse margin is minimized (= the margin is maximized) over both the classifier's weights and the embedding; shouldn't it be "the best linear classifier"? ============================ I have read the rebuttal, which clarified all the issues raised in my review. Thank you.

Confidence in this Review

1-Less confident (might not have understood significant parts)


Reviewer 4

Summary

The binary matrix completion problem asks that given a binary matrix and a process that each turn produces a randomly chosen entry of this matrix, find a strategy to geuss the entries of the matrix with as few as possible mistakes. The authors adapt the matrix exponential gradient algorithm in order to solve this problem. They bound the number of the mistakes as a function of margin complexity of the binary matrix and its dimension. As the margin complexity is bounded from above by (a function of) the trace norm of the matrix U, it produce un upper bound as a function of the trace norm. While there are known upper bounds as a function of the trace norm using other algorithms which are better than this bound, it is important to note that the margin complexity can be much smaller than the trace norm, thus this paper's algorithm may provide a much better bound. Finally, the authors show that the number of mistakes their algorithm makes is at most a logarithmic factor from the number of mistakes made by the Kernel Perceptron with an optimal kernel in hindsight.

Qualitative Assessment

The problem of matrix completion seems to be an interesting and important problem, and finding an upper bound using the margin complexity instead of the trace norm appears to be even more important as it is a very natural property and in general can be much smaller than the trace norm. At some points, the paper is written in a confusing manner. For example, in lines 47-49, the authors mention that their result have a "more limited applicability" on the one hand, and on the other it become "non trivial" after fewer observed matrix entries. Both points here are not clear and seem to be important. It might also be preferable if the authors mention in this paragraph that all trace norm, margin complexity and gamma_2 norm will be defined soon, as these are less known norms of matrices. Another example is corollary 3.2. It is said "for all gamma" and then in the next line gamma=1/k. This is a very confusing writing of an important corollary which let us avoid choosing a learning rate. Couldn't understand the meaning of line 161-162. The entire paragraph in lines 175-184 uses references in a confusing way. It keeps referencing [16] and [17] and expects the reader to remember which is which. There should be some better way to write about these results (name of authors or of the algorithm?). This paragraph and the one before it also give the main reason why margin complexity is better than the trace norm. It might be preferable if some of these details will more to the introduction. Also, while in general the margin complexity can be much smaller (as the authors give such an example), it is very interesting to know if this is likely the case, namely how many of the binary matrices of a fixed (large) dimension have very small margin complexity in compared to the trace norm. Two minor comments line 80 - the definition of SP(U) should contain "for all i,j" line 215 - do you mean "exactly r" or "at least r"?

Confidence in this Review

1-Less confident (might not have understood significant parts)


Reviewer 5

Summary

The paper studies an online setting in which there is an unknown binary matrix and at each stage the learner is asked to predict a particular entry, then given the true entry. The authors propose an online algorithm for predicting the binary matrix, based upon the matrix exponentiated gradient algorithm of Tsuda et al. The principle theorem is an upper bound on the number of mistakes made by the aforementioned algorithm. This bound is based upon the notion margin complexity of sequences, which extends the notion of margin complexity of a matrix. The bound is also proven to be tight as a function the margin complexity, up to a poly-logarithmic factor. An impressive feature of the bound is that equivalent up to a logarithmic factor to the bound on the kernel perceptron as function of a margin, even though the optimal kernel is not known in advance. The authors also derive a bound on the margin complexity of a bi-clustered matrix, which is combined with the main theorem to to show that the main algorithm performs bi-clustering with a bound on the number of mistakes made. This bound relates favourably to other bi-clustering algorithms in the literature.

Qualitative Assessment

The paper makes an important contribution to the field of online learning. It is both novel and technically strong. A key novel feature of the paper is that the learner operates in a paradigm in which the graph structure is never revealed to the learner. There are a range of technical contributions in this novel scenario, including an algorithm, a mistake bound on the algorithm, and a demonstration that the bound is tight, up to a poly-logarithmic factor. There are also insightful connections both to the kernel perceptron algorithm and to the field of bi-clustering. Whilst the work is theoretical, the connections to both multi-task learning and bi-clustering, mean that the algorithm has a potential for impact at the more applied end of the machine learning spectrum. Overall the paper is very well presented. There are a few potential areas for improvement in this regard. Firstly, I think the authors should highlight the fact that the upper bound (Theorem 3.1) is optimal up to a poly-logarithmic factor (ie. Theorem 5.4) either just before, or just after the statement of Theorem 3.1. Secondly, the second part of Section 4 is slightly unclear. I think there is also a typo in equation (6) following line 226, where the inequality sign is in the wrong direction. Thirdly, Section 2 might be improved by giving an intuitive description of the notion of margin complexity for matrices. Finally, it would be very interesting to see how the algorithm behaves empirically on some artificial or real-world data sets, and observe how tight the bound is in practice. That said, perhaps this is not possible in the limited space available. Overall, the paper was extremely insightful and a pleasure to read.

Confidence in this Review

2-Confident (read it all; understood it all reasonably well)


Reviewer 6

Summary

This paper uses an online matrix multiplicative weights algorithm for online binary matrix completion. Different from existing results, this work provides an error bound measured in the margin complexity of the matrix but under stronger constraints. Comparison and relation of the work with other existing results are well discussed in the paper.

Qualitative Assessment

While the techniques in this paper is similar to some existing results in the literature, it is applied to a new problem setting. The paper is well written and easy to follow. One point that I am concerned is the impact of the result. 1. The problem setting is motivated in Section 1, linked to the online prediction of a set of objects. I am not familiar with this topics. How is the result of Theorem 3.1, when applied to the online prediction of set of objects, compared to the existing results? 2. The discussion in Line 158- Line 184, and that in Line 252 - Line 257 is really appreciated. But on the other hand, based on these discussions, the result of this paper seems to have limited application. 3. The main result of this paper, Theorem 3.1, requires the existence of a consistent predictor. This condition could be a strong condition. It may needs more discussion.

Confidence in this Review

1-Less confident (might not have understood significant parts)